# Fabrication of Simultaneously Implementing “Wired Face-Up and Face-Down Ultrathin Piezoresistive Si Chips” on a Film Substrate by Screen-Offset Printing

**DOI:** 10.3390/mi10090563

**Published:** 2019-08-26

**Authors:** Yusuke Takei, Ken-ichi Nomura, Yoshinori Horii, Daniel Zymelka, Hirobumi Ushijima, Takeshi Kobayashi

**Affiliations:** Sensing System Research Center, National Institute of Advanced Industrial Science and Technology (AIST), Tsukuba 305-8564, Japan

**Keywords:** screen-offset printing, piezoresistive Si, blood pulse measurement, flexible hybrid electronics

## Abstract

We realized the implementation of an ultrathin piezoresistive Si chip and stretchable printed wires on a flexible film substrate using simple screen-offset printing technology. This process does not require a special MEMS fabrication equipment and is applicable to face-up chips where electrodes are formed on the top surface of the chip, as well as to face-down chips where electrodes are formed on the bottom surface of the chip. This fabrication process is quite useful in the field of flexible hybrid electronics (FHE) as a method for mounting and wiring electronic components on a flexible substrate. In this study, we confirmed that face-up and face-down chips could be mounted on polyimide film tape. Furthermore, it was confirmed that the two types of chips could be simultaneously mounted even if they exist on the same substrate. Five-μm-thick piezoresistive Si chips were transferred and wired on a polyimide film tape using screen-offset printing, and a band-plaster type blood pulse sensor was fabricated. Moreover, we successfully demonstrated that the blood pulse could be measured with neck, inner elbow, wrist, and ankle.

## 1. Introduction

Recently, researches on flexible hybrid electronics (FHE) have been actively studied [1,2,3,4,5,6,7,8]. Moreover, ultrathin Si-based devices/chips with a thickness of 50 μm or less, which do not break even when bent, are mounted on a flexible substrate to realize flexible and high-sensitivity sensors. Based on this technology, various flexible and high-sensitivity sensors are being developed [9,10,11,12,13,14].

For example, Harendt et al. realized hybrid systems in foil (HySiF), which implemented a 10-μm-thick face-down chip on foil and then formed by laser drilling and refilled by a metal to make electrical contact between chip and wiring [2]. Kim et al. realized flexible Si NAND flash memory by implementing a 340-nm-thick face-down chip on a bump, which formed on the flexible substrate [5]. These researches target face-down chip, where the electrodes are on the bottom surface of the chip. However, there is another type of chip—a “face-up chip”, in which electrodes are on the top surface of the chip. Therefore, we aimed to develop and implement a method applicable to flexible hybrid electronics that can handle both “face-up” and “face-down” chip.

To address this issue, we propose a new implementing process that can integrally mount and wire on a flexible substrate even if face-up and face-down chips simultaneously exist on the same wafer.

This process uses a screen-offset printing technique [15,16,17,18,19], in which the ultrathin Si chip together with wirings printed with the stretchable conductive ink is transferred to an adhesive film via a blanket made of silicone rubber. Face-up and face-down devices/chips can be mounted on a flexible substrate without using an expensive MEMS mounting machine, and wirings can also be formed (Figure 1).

## 2. Fabrication

### 2.1. Chip Implementation Using Screen-Offset Printing

In this study, screen-offset printing was used to mount the chips and wires on the film substrate. As shown in Figure 2a, screen printing involves printing by applying ink to the object through the holes in the mask. On the other hand, in screen offset printing, as shown in Figure 2b, the ink that has passed through the holes in the mask is temporarily transferred to the blanket roll, and printing is performed by pressing the blanket roll against the object.

Screen printing has a problem that the ink spreads after printing, and thus this printing method is not suitable for printing a fine pattern. However, in screen-offset printing, once the ink is printed on a blanket roll made of silicone rubber, the solvent in ink is absorbed by the silicone rubber, and the spreading of the ink after printing is suppressed. Therefore, screen-offset printing enables printing of finer patterns than screen printing.

In this research, in screen offset printing, we considered that the surface of the printed ink was reversed, since the ink once transferred to the blanket roll. Therefore, we considered that screen-offset printing is similar to a flip chip bonder, and by combining screen printing and screen-offset printing, we can achieve the implementing process, which can treat face-up and face-down chips simultaneously.

### 2.2. Fabrication of Ultrathin Face-Up/Face-Down Chip

In this study, in order to demonstrate the proposed mounting process, we fabricated ultrathin chips with face-up and face-down electrodes, which does not break even if mounted on a flexible substrate. The fabrication process of the ultrathin piezoresistive silicon chip is shown in Figure 3a [20].

First, we fabricated 5-μm-thick ultrathin piezoresistive silicon membrane from the 5-μm-thick device layer of silicon on insulator (SOI) wafer (5/2/500 μm). To form a piezoresistive layer on the surface of the device-layer silicon (P-type), phosphorus ion dopant (OCD P-59230, Kogyo CO., Ltd., Tokyo, Japan) was spin-coated, and thermal diffusion was performed using an annealing furnace. After that, a gold electrode was formed, and the device-layer silicon was etched by Deep Reactive Ion Etching (DRIE) to form an ultrathin piezoresistive silicon membrane (Figure 3a(1)). The membrane has a rectangular shape of 1 mm × 5 mm. Then, the handling-layer silicon was patterned and etched from the back (Figure 3a(2)), so that 5-μm-thick ultrathin piezoresistive silicon membranes supported by six beams were fabricated, as shown in Figure 3b,c. The resistance between two au electrodes was 1.2 kΩ.

### 2.3. Implementing Process for Face-Up chip

In this section, we describe the implementing method of the fabricated ultrathin piezoresistive Si chip onto the flexible substrate with its electrode on face-up position.

Figure 4 shows a process outline of transferring the face-up ultrathin Si chip and wiring on a polyimide film tape using screen-offset printing. First, the chip is transferred onto the polyimide film tape (Tesa, model 51408, width: 12 mm) by using the Polydimethylsiloxane (PDMS) blanket (Figure 4(1)). As a result, as shown in Figure 4(2), the Si chip is transferred to the PDMS blanket with the electrode facing downward. After that, as shown in Figure 4(3), the Si chip transferred onto the PDMS blanket is peeled off with a polyimide film tape. Then the electrode is transferred onto the polyimide film tape with the electrode facing upward. Then, as shown in Figure 4(4), a screen offset printing machine is used to print wiring with silver ink (TOYOBO, SSP2801, Osaka, Japan) on the face-up Si chip transferred onto the polyimide film tape. The diameter of the blanket roll is 151.4 mm, and for alignment, we first printed Ag ink on a polyethylene terephthalate film and checked where Ag ink was printed using cameras. Then, according to the information from the cameras, the position of a substrate with Si chips was adjusted so that their electrodes could be covered with Ag ink. After the screen-offset printing, the printed silver ink is baked at 150 °C for 30 min to complete the sintering (Figure 4(5)). The volume resistivity of the printed Ag ink was 2.03 × 10^−5^ Ω·cm. As shown in the photograph of Figure 4(5), we confirmed that the Si chip was transferred on the polyimide film tape in the face-up position, and the printed wiring is formed of silver ink to contact the electrodes at both ends of the chip.

### 2.4. Implementing Process for Face-Down Chip

Next, we describe how to mount the face-down chip. Figure 5 shows the process outline. First, as shown in Figure 5(1), we place the ultra-thin piezoresistive Si supported by a thin beam from the frame, fabricated in Section 2.2, on the PDMS bracket. Then, the chip is punched out from above with a collet and pressed against the PDMS blanket, as shown in Figure 5(2), and the electrode is transferred in a face-up state on the PDMS blanket. Next, a screen offset printing machine is used to form a wire with silver ink to be in contact with the electrodes at both ends of the ultrathin piezoresistive Si element (Figure 5(3)). After baking the ink at 150 °C for 30 min (Figure 5(4)), we peel off the wiring and the Si chip on the adhesive surface of the polyimide film tape as shown in Figure 5(5). As a result, as shown in Figure 5(6), the ultra-thin Si chip wired in the face-down state is successfully transferred onto the polyimide film tape.

### 2.5. Implementing Process for a Mixture of Face-Up and Face-Down Chip

Finally, we will describe the mounting method when the face-up chip and the face-up chip exist together on the same surface. Figure 6 outlines the process. This process, shown in Figure 6a(1), starts from the situation where face-up chip and face-down chip are transferred side by side on the PDMS blanket.

First, wiring is printed with silver ink by screen printing process on the electrodes at both ends of the chip with the electrodes facing upward (Figure 6a(2)).

After that, the adhesive surface of the polyimide film tape is brought into contact from the top, and the chip and the wiring as described above are peeled off with the adhesive layer of the tape (Figure 6a(3,4)). Then, the face-down chip wired by printed wiring and the face-up chip not yet wired is transferred onto the polyimide film tape (Figure 6a(5)). Then, wiring is formed by screen offset printing on the electrodes on the surface of the face-up chip not yet wired. Then, the ink is baked to complete sintering (Figure 6a(6)). Figure 6b shows a face-up chip and a face-down chip mounted on polyimide film tape.

## 3. Film-Type Blood Pulse Sensor Demonstration

In order to demonstrate the advantage of our method, in which we can mount the ultra-thin piezoresistive Si chip on adhesive surface of polyimide film tape, we have developed a film-type blood pulse sensor that can easy attach to the skin due to the adhesive surface of a polyimide film tape. The sensor was using a face-up ultra-thin piezoresistive Si chip mounted on a polyimide film tape fabricated by the process shown in Figure 4.

Before measurement, we evaluated the piezoresistivity of the fabricated piezoresistive Si chip transferred on the polyimide film tape. Piezoresistivity was evaluated by fixing the fabricated sensor and a commercially available strain gauge (Kyowa Electronic Instruments, KFRB-5-120-C1-11, gauge factor: 2, Tokyo, Japan) in parallel to an aluminum plate and comparing the response to bending. Figure 7a shows the result. It is found that the piezoresistivity of this sensor is strongly influenced by the properties of the underlying polyimide film, and thus, this sensor has a large hysteresis. From the measured result shown in Figure 7a, we calculated the gauge factor of the sensor as 2.98. Figure 7b shows the measuring setup for the sensor. Figure 8 shows the measurement results of blood pulses with the fabricated film type sensor at the neck, inner elbow, wrist, and ankle. Since the piezoresistive Si chip transferred to the adhesive surface of the polyimide film is as thin as 5 μm and does not break even when bent, it can be easily attached to the skin and can tightly fit the curved surface. As a result, we confirm that the characteristics of the blood pulse waveform corresponding to the distance from the heart can be measured. In particular, two peaks (First peak and second peak) were observed on the wrist, and three peaks (first peak, second peak, and diastolic peak) were clearly measured at neck and ankle.

## 4. Conclusions

We propose the implementation method of an ultrathin piezoresistive Si chip and stretchable printed wires on a flexible film substrate by the combination of screen printing and screen-offset printing. This process does not require special MEMS fabrication equipment and is applicable to face-up chips where electrodes are formed on the top surface of the chip, as well as to face-down chips where electrodes are formed on the bottom surface of the chip. Furthermore, it was confirmed that the two types of chips could be simultaneously mounted even if they exist on the same substrate. Five-μm-thick piezoresistive Si chips were transferred and wired on a polyimide film tape using screen-offset printing, and we confirmed that we can measure the blood pulse with the 5-μm-thick piezoresistive Si chip mounted and implemented on polyimide film tape. Our proposed method is useful in the field of flexible hybrid electronics as a method for mounting and wiring electronic components on a flexible substrate.

## Figures and Tables

**Figure 1 micromachines-10-00563-f001:**
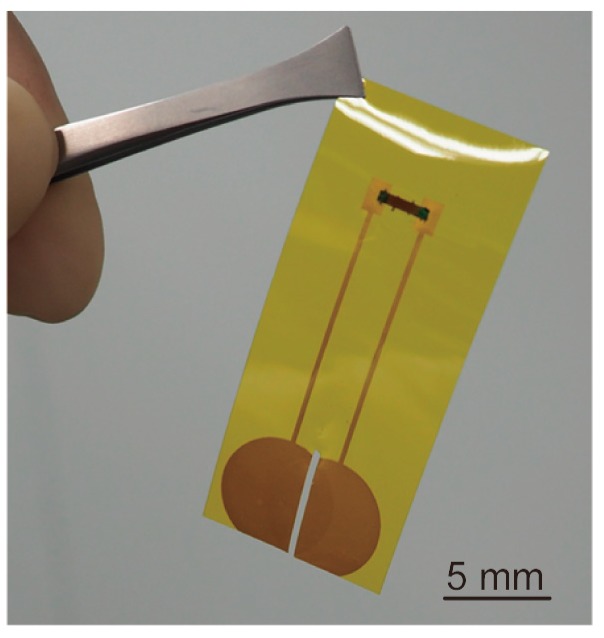
Photograph of the device that was fabricated by the proposed method. An ultrathin piezoresistive Si chip with a thickness of 5 μm was mounted on the surface of the polyimide film tape together with the printed wirings.

**Figure 2 micromachines-10-00563-f002:**
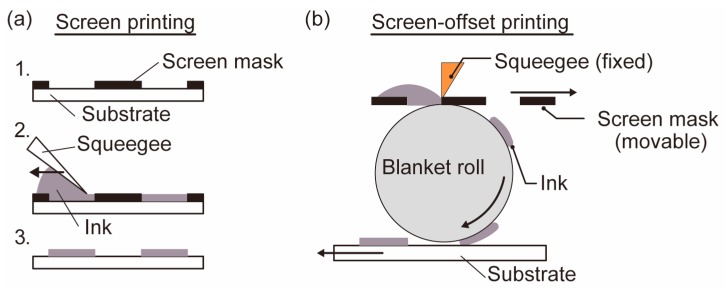
The process outline of (**a**) screen printing and (**b**) screen offset printing.

**Figure 3 micromachines-10-00563-f003:**
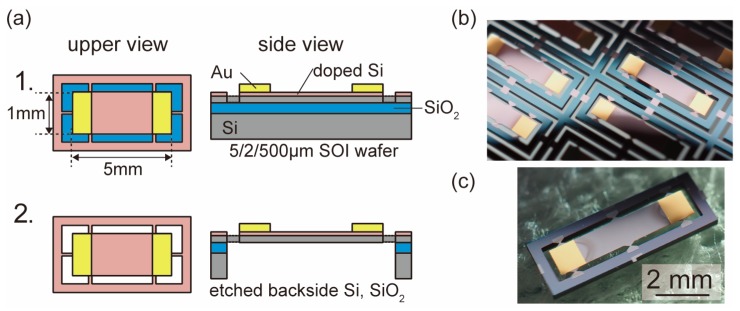
(**a**) Fabrication of ultrathin piezoresistive Si chip. 5-μm-thick ultrathin piezoresistive Si chip was fabricated from the device layer of 5/2/500 μm silicon on insulator (SOI) wafer. (**b**) A fabricated chip on the wafer. (**c**) Ultrathin Si chip was supported by six beams.

**Figure 4 micromachines-10-00563-f004:**
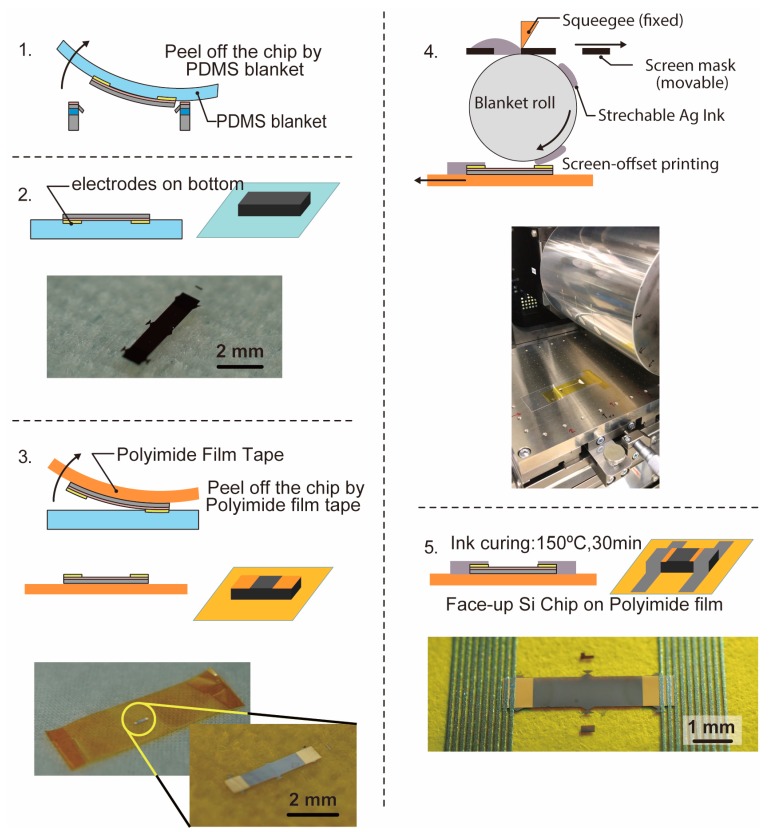
Process outline of transferring the face-up ultrathin Si chip and wiring on a polyimide film tape using screen-offset printing.

**Figure 5 micromachines-10-00563-f005:**
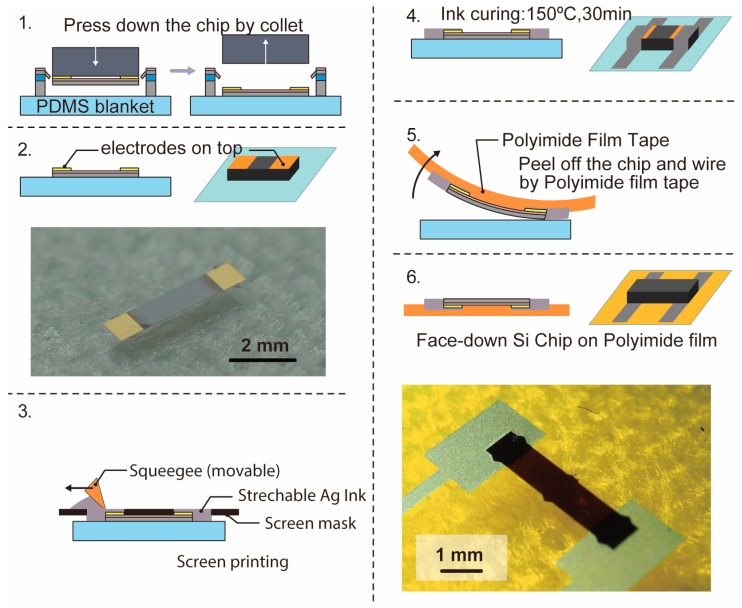
Process outline of transferring the face-down ultrathin Si chip and wiring on a polyimide film tape.

**Figure 6 micromachines-10-00563-f006:**
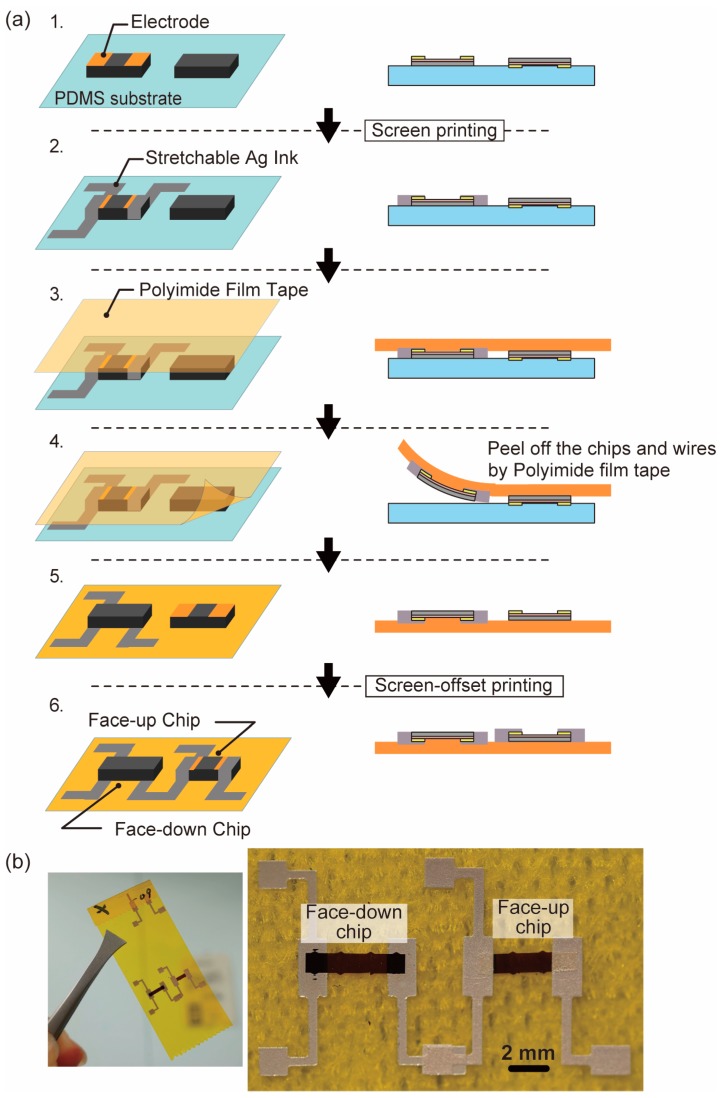
(**a**) Chip transfer and screen-offset printed wiring process of face-up and face-down chip mixture situation. (**b**) Photograph of the implemented face-up and face-down 5-μm-thick piezoresistive Si chips on polyimide film tape.

**Figure 7 micromachines-10-00563-f007:**
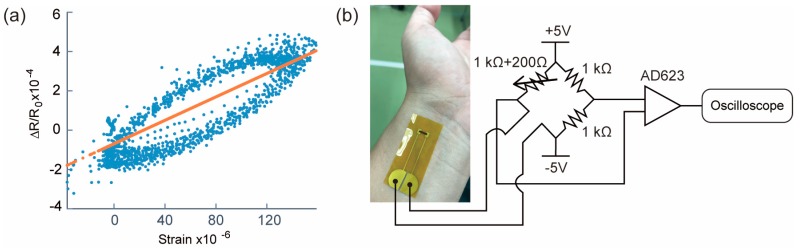
(**a**) Relationship between strain and the relative resistance change of the ultrathin piezoresistive Si on film. (**b**) Schematic of the blood pulse measuring setup.

**Figure 8 micromachines-10-00563-f008:**
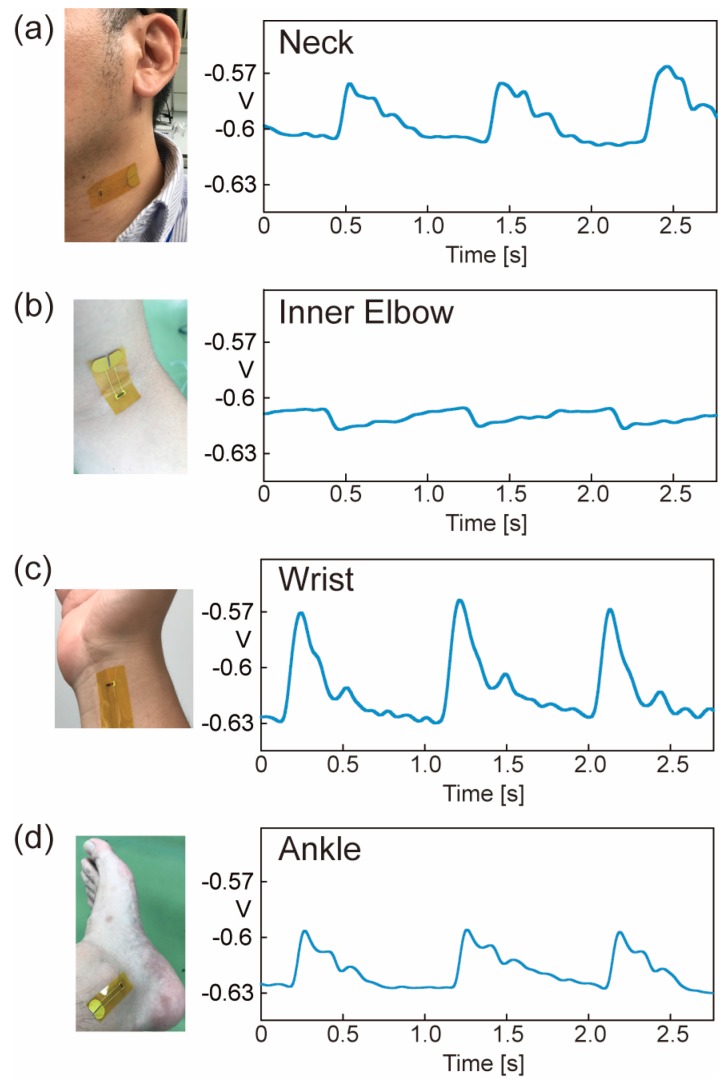
Demonstration to measure blood pulse with our fabricated film-type sensor at (**a**) neck, (**b**) inner elbow, (**c**) wrist, and (**d**) ankle.

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
