# Peer review of "Fabrication of Simultaneously Implementing “Wired Face-Up and Face-Down Ultrathin Piezoresistive Si Chips” on a Film Substrate by Screen-Offset Printing"

_micromachines, 2019, doi:10.3390/mi10090563_

Round 1

Reviewer 1 Report

- It is necessary to elaborate on the examination of the other previous researches in the introduction.To show the originality of paper clearly to describe the difference or advance of this paper in comparison with previous papers.

- The manufacturing process of the blood pressure sensor is not clearly described. Whether the blood pressure sensor is a proper application that shows the feasibility and the usefulness of the proposed process which simultaneously produces the face-up and face-down flexible chips or not.

 > It is desirable to present the physical results(including pictures and drawings) of each process in conjunction with manufacturing process of the blood pressure sensor.

- The conclusion is too shot to summarize the paper. 

Author Response

We appreciate the careful reading and valuable suggestions of the reviewer which allowed us improving our manuscript. We have carefully considered the comments and have revised the manuscript as summarized below.

Comment 1

- It is necessary to elaborate on the examination of the other previous researches in the introduction. To show the originality of paper clearly to describe the difference or advance of this paper in comparison with previous papers.

Answer to Comment 1

We thank the reviewer for the comments which help us to significantly improve the paper. We revised the Introduction as follows.

Before:(Line 29)

Recently, researches on Flexible Hybrid Electronics (FHE) have been actively studied [1-8]. Moreover, ultrathin Si-based devices/chips with a thickness of 50μm or less, which do not break even when bent, are mounted on a flexible substrate to realize flexible and high sensitive sensors. Based on this technology, various flexible and high sensitivity sensors are being developed [9-14].

However, there are two types of chips: a “Face-up chip” which electrodes are on the top surface of the chip, and a “Face-down chip” which electrodes are on the bottom surface of the chip. Therefore, it is necessary to select the mounting and wiring methods properly, depending on these chip types, which causes a problem that the mounting and wiring processes become complicated. To address this issue, we propose a new implementing process that can integrally mount and wire on a flexible substrate even if Face-up and Face-down chips simultaneously exist on the same wafer.

This process uses a screen-offset printing technique [15-19], in which the ultrathin Si chip together with wirings printed with the stretchable conductive ink is transferred to an adhesive film via a blanket made of silicone rubber. Face-up and Face-down devices/chips can be mounted on a flexible substrate without using an expensive MEMS mounting machine, and wirings can also be formed (Figure 1).

After: (Line 29)

Recently, researches on Flexible Hybrid Electronics (FHE) have been actively studied [1-8]. Moreover, ultrathin Si-based devices/chips with a thickness of 50μm or less, which do not break even when bent, are mounted on a flexible substrate to realize flexible and high sensitive sensors. Based on this technology, various flexible and high sensitivity sensors are being developed [9-14].

For example, Harendt et al., realized Hybrid Systems in Foil (HySiF), which implementing 10 μm-thick Face-down chip on a foil and then forming via by laser drilling and refill the via by a metal to make electrical contact between chip and wiring [2]. Kim et al., realized flexible Si NAND flash memory by implementing 340 nm-thick Face-down chip on a bump which formed on the flexible substrate [5]. These researches target Face-down chip, where the electrodes are on the bottom surface of the chip. However, there is another type of chip: a “Face-up chip” which electrodes are on the top surface of the chip. Therefore, we aimed to develop a implementing method applicable to Flexible Hybrid Electronics that can handle both “Face-up” and “Face-down” chip.

To address this issue, we propose a new implementing process that can integrally mount and wire on a flexible substrate even if Face-up and Face-down chips simultaneously exist on the same wafer. This process uses a screen-offset printing technique [15-19], in which the ultrathin Si chip together with wirings printed with the stretchable conductive ink is transferred to an adhesive film via a blanket made of silicone rubber. Face-up and Face-down devices/chips can be mounted on a flexible substrate without using an expensive MEMS mounting machine, and wirings can also be formed (Figure 1).

Comment 2

- The manufacturing process of the blood pressure sensor is not clearly described. Whether the blood pressure sensor is a proper application that shows the feasibility and the usefulness of the proposed process which simultaneously produces the face-up and face-down flexible chips or not.

Answer to Comment 2

The blood pulse sensor in this paper is a Face-up chip mounted on an adhesive layer of polyimide film tape. This demonstration is not intended to show the advantage of being able to mount both Face-up and Face-down chips, but to show the advantage of mounting a ultrathin semiconductor element on the adhesive layer of film tape. Therefore, we revised as follows to clarify the intention of this demonstration.

Before:

As an application of this implementation method, a film-type blood pulse sensor was fabricated.

After:(Line154)

In order to demonstrate the advantage of our method, which we can mount the ultra-thin piezoresistive Si chip on adhesive surface of polyimide film tape, we have developed a film-type blood pulse sensor that can easy attach to the skin due to the adhesive surface of a polyimide film tape. The sensor was using a Face-up ultra-thin piezoresistive Si chip mounted on a polyimide film tape fabricated by the process shown in Figure 4.

Comment 3

- It is desirable to present the physical results (including pictures and drawings) of each process in conjunction with manufacturing process of the blood pressure sensor.

Answer to Comment 3

The sensor was fabricated by the process shown in Figure 4. So We revised the manuscript as follows.

After:(Line 157)

The sensor was using a Face-up ultra-thin piezoresistive Si chip mounted on a polyimide film tape fabricated by the process shown in Figure 4.

Comment 4

- The conclusion is too shot to summarize the paper. 

Answer to Comment 4

We revised the conclusion as follows.

Before:

We proposed the process to implement “wired face-up and face-down chips” on a film substrate. This process is realized by a simple screen-offset printing technique, so it does not require special MEMS fabrication equipment. As a demonstration, we fabricated a film-type blood pulse measurement sensor, which 5μm-thick piezoresistive Si chips were transferred and wired on a polyimide film tape. Moreover, we confirmed that the blood pulse could be measured at the neck and wrist.

After:

We proposed the implementation method of an ultrathin piezoresistive Si chip and stretchable printed wires on a flexible film substrate by the combination of screen printing and screen-offset printing. This process does not require a special MEMS fabrication equipment and applicable to Face-up chips where electrodes are formed on the top surface of the chip, as well as to Face-down chips where electrodes are formed on the bottom surface of the chip. Furthermore, it was confirmed that the two types of chips could be simultaneously mounted even if they exist on the same substrate. As its application, 5 μm-thick piezoresistive Si chips were transferred and wired on a polyimide film tape using screen-offset printing. And we confirmed that we can measure the blood pulse with the 5 μm-thick piezoresistive Si chip mounted implemented on polyimide film tape. Our proposed method is useful in the field of Flexible Hybrid Electronics as a method for mounting and wiring electronic components on a flexible substrate.

Reviewer 2 Report

The authors present a method to establish electrical contact to ultra-thin silicon devices by screen printing and screen-offset printing. The proposed method allows top and bottom as well as two-sided contacting. The functionality of the method is proven by fabricating a pulse measurement device comprised of an ultra-thin, highly sensitive Si-piezoresistor mounted on polyimide tape and electrically contacted with the presented methods.

The overall quality of the paper is high. I feel that assembly and mounting technology is underinvestigated and, thus, appreciates the presented approach and implementation. Many recently developed devices suffer of a lack of advanced and suitable mounting technologies.

The paper is easy to read and understand. There are minor grammar mistakes, which I suggest to be corrected. However, the mistakes do not confine the understandability. The figures are easy to understand and support the text very well. The captions of fig. 5, fig. 7 and fig. 8 seem to be incorrect.

The description of the implementation of the proposed method is generally sufficient. Some details may be added to the description, though, e.g. precise type and relevant properties of polyimide tape, precise type of silver ink, diameter/radius of blanket roll. Additionally, please add information regarding

- alignment of the Si-chip/Polyimide tape for printing

- minimum bending radius before cracking of the final device (experimental results or at least an estimation)

- resulting absolute resistance of the Si-piezoresistor (chip only) and the printed tracks (contact pad to chip)

- gauge factor/strain sensitivity of the printed tracks which may falsify measurement results (only in case the absolute resistance of the tracks is non-negligible)

gauge factor of the final device (may be estimated from fig. 7 results)

- details on the evaluation method for strain sensitivity

The demonstration of the device performance as pulse sensor is impressive. As the MDPI Micromachines journal is focused on fabrication, I do not strictly require the authors to add additional results. However, I suggest to add some details on the bridge circuit setup such as excitation voltage. Additionally, it would be nice to add a reference signal of a established pulse measurement method (e.g. pulse oximetry).

In the conclusion part, please add information on possible applications for the method and an assessment on limitations that you see in the application (e.g. maximum Si-chip height, minimum structures sizes achieveable, mechanical stability).

Author Response

We appreciate the careful reading and valuable suggestions of the reviewer which allowed us improving our manuscript. We have carefully considered the comments and have revised the manuscript as summarized below.

Comment 1

There are minor grammar mistakes, which I suggest to be corrected. However, the mistakes do not confine the understandability.

Answer 1

Thank the reviewer for kind comment. We will use the English proofreading service provided by the journal.

Comment 2

The captions of fig. 5, fig. 7 and fig. 8 seem to be incorrect.

Answer 2

Thank you for the comment. The captions in Figs. 5, 7, and 8 have been revised as follows.

Figure 5. Process outline of transferring the Face-down ultrathin Si chip and wiring on a polyimide film tape. using screen-offset printingDetail of the ultrathin Si chip transfer and screen-offset printing process. (a) Process for Face-up type chip. (b) Process for Face-down type chip. Both type chips were successfully transferred and wired on polyimide film tape.

Figure 7. (a) Relationship between strain and the relative resistance change of the ultrathin piezoresistive Si on film. (b)(c) Demonstration to measure blood pulse with our fabricated film-type sensor. Schematic of the blood pulse measuring setup.

Figure 8. (a) Relationship between strain and the relative resistance change of the ultrathin piezoresistive Si on film. (b)(c) Demonstration to measure blood pulse with our fabricated film-type sensor.  at (a) neck, (b) inner elbow, (c) wrist, and (d) ankle.

Comment 3

Some details may be added to the description, though, e.g. precise type and relevant properties of polyimide tape, precise type of silver ink, diameter/radius of blanket roll.

Answer 3

Thank the reviewer for the comment. We add the description as follows.

(Line 98)

First, the chip is transferred onto the polyimide film tape (Tesa, model 51408, width:12mm) by using the PDMS blanket (Figure 4-1).

(Line 103)

Then, as shown in Figure 4-4, a screen offset printing machine is used to print wiring with silver ink (TOYOBO, SSP2801) on the Face-up Si chip transferred onto the polyimide film tape.

(Line 105)

The diameter of the PDMS blanket is 151.4 mm.

Comment 4

Additionally, please add information regarding

- alignment of the Si-chip/Polyimide tape for printing

Answer 4

Thank the reviewer for the comment. We add the description of the alignment as follows.

(Line 105)

And for alignment, we first printed Ag ink on a polyethylene terephthalate film, and checked where Ag ink was printed using cameras. Then, according to the information from the cameras, the position of a substrate with Si chips was adjusted so that their electrodes could be covered with Ag ink.

Comment 5

- minimum bending radius before cracking of the final device (experimental results or at least an estimation)

Answer 5

Empirically, we confirmed that in the case of ultra-thin Si with a thickness of 5µm, the final device does not break up to a bending radius of 5 mm. However, since we have not conducted the experiment with various thicknesses of Si, we would like it to be a future work.

Comment 6

- resulting absolute resistance of the Si-piezoresistor (chip only) and the printed tracks (contact pad to chip)

Answer 6

We add the description as follows.

(Line 89)

The resistance between two au electrodes was 1.2 kohm.

(Line 110)

The volume resistivity of the printed Ag ink was 2.03 × 10-5 ohm⋅cm.

Comment 7

- gauge factor/strain sensitivity of the printed tracks which may falsify measurement results (only in case the absolute resistance of the tracks is non-negligible)

Answer 7

The resistance value of the wiring part is sufficiently small compared to the resistance value of the Si piezoresistive element, so it can be ignored.

Comment 8

- gauge factor of the final device (may be estimated from fig. 7 results)

Answer 8

As a result of calculating the gauge factor from the experimental results in Figure 7, the gauge factor of the final device is 2.98. We revised the manuscript as follows.

(Line 165)

From the measured result shown in Figure 7(a), we calucurated the gauge factor of the sensor as 2.98.

Comment 9

- details on the evaluation method for strain sensitivity

Answer 9

We added the description as follows.

(Line 161)

Piezoresistivity was evaluated by fixing the fabricated sensor and a commercially available strain gauge (Kyowa Electronic Instruments, KFRB-5-120-C1-11, gauge factor:2) in parallel to an aluminum plate, and comparing the response to bending.

Comment 10

I suggest to add some details on the bridge circuit setup such as excitation voltage.

Answer 10

Thank the reviewer for the comment. We added the figure of the bridge circuit in Figure 7(b).

Comment 11

Additionally, it would be nice to add a reference signal of an established pulse measurement method (e.g. pulse oximetry).

Answer 11

Thank you for the comment. But unfortunately, we didn’t measure the reference data at the experiment, we would like to be future work.

Comment 12

In the conclusion part, please add information on possible applications for the method and an assessment on limitations that you see in the application (e.g. maximum Si-chip height, minimum structures sizes achievable, mechanical stability).

Answer 12

Thank you for the comment. The Coclusion part has been modified as follows.

Before:

We proposed the process to implement “wired face-up and face-down chips” on a film substrate. This process is realized by a simple screen-offset printing technique, so it does not require special MEMS fabrication equipment. As a demonstration, we fabricated a film-type blood pulse measurement sensor, which 5μm-thick piezoresistive Si chips were transferred and wired on a polyimide film tape. Moreover, we confirmed that the blood pulse could be measured at the neck and wrist.

After: (Line 186)

We proposed the implementation method of an ultrathin piezoresistive Si chip and stretchable printed wires on a flexible film substrate by the combination of screen printing and screen-offset printing. This process does not require a special MEMS fabrication equipment and applicable to Face-up chips where electrodes are formed on the top surface of the chip, as well as to Face-down chips where electrodes are formed on the bottom surface of the chip. Furthermore, it was confirmed that the two types of chips could be simultaneously mounted even if they exist on the same substrate. As its application, 5 μm-thick piezoresistive Si chips were transferred and wired on a polyimide film tape using screen-offset printing. And we confirmed that we can measure the blood pulse with the 5 μm-thick piezoresistive Si chip mounted implemented on polyimide film tape. Our proposed method is useful in the field of Flexible Hybrid Electronics as a method for mounting and wiring electronic components on a flexible substrate.

Reviewer 3 Report

Fabrication of Simultaneously Implementing “Wired Face-up and Face-down Ultrathin Piezoresistive Si Chips” on a Film Substrate by Screen-offset Printing  

Yusuke Takei, Ken-ichi Nomura, Yoshinori Horii, Daniel Zymelka, Hirobumi Ushijima, and Takeshi Kobayashi

Herewith I am submitting my reviewer comments for the above-mentioned manuscript which is under consideration to be published in Micromachines. In their article, the authors fabricated a thin piezoresistive Si chip and stretchable  printed wires on a flexible film substrate where one can use face down and face up techniques on the same device. The paper is a very technical one. While the technology sounds quite innocent I am sure quite some engineering went into the process. They used their films to sense the blood pulse. I am not too familiar with this kind of fabrication techniques but to me it looks quite intriguing. It sounds simple enough that it might actually be implemented in practice. So overall I quite liked the paper. The article is well-written and the figures nicely explain how their process works.

Line 86: “At this section, we will describe the” should be “In this section, we will describe the”

Line 92: “Then, this time, the electrode is transferred” should be “Then, the electrode is transferred”

Line 95: “After that, the printed silver nano ink is baked at 150 ° C. for 30 minutes to complete” to complete what? There is sthg missing in the sentence

Line 104: “First, as shown in Figure 5-1, place the ultra-thin piezoresistive” here the text changes into imperative. Please be consistent with the tense.

Line 119: “In this process, as shown in Figure 6 (a)- 1, it starts from the situation where Face-up” should be “This process, shown in Figure 6 (a)- 1, starts from the situation where Face-up”

Line 128: “Then, the ink is sintered to complete” to complete what? There is sthg missing in the sentence

Figure 8: I think it would be nice to say a few words on how these can be interpreted. What do you look at? The distance between peaks? The signal height? And what does it mean?

Author Response

We appreciate the careful reading and valuable suggestions of the reviewer which allowed us improving our manuscript. We have carefully considered the comments and have revised the manuscript as summarized below.

Comment 1

Line 86: “At this section, we will describe the” should be “In this section, we will describe the”

>> Thank you for the suggestion. We revised as mentioned.

Comment 2

Line 92: “Then, this time, the electrode is transferred” should be “Then, the electrode is transferred”

>> Thank you for the suggestion. We revised as mentioned.

Comment 3

Line 95: “After that, the printed silver nano ink is baked at 150 ° C. for 30 minutes to complete” to complete what? There is sthg missing in the sentence

>> Thank you for the suggestion. We revised the manuscript as follows.

(Line 108)

After the screen-offset printing, the printed silver ink is baked at 150 ° C for 30 minutes to complete the sintering (Figure 4-5).

Comment 4

Line 104: “First, as shown in Figure 5-1, place the ultra-thin piezoresistive” here the text changes into imperative. Please be consistent with the tense.

>> Thank you for the suggestion. We revised the manuscript as follows.

First, as shown in Figure 5-1, we place the ultra-thin piezoresistive Si supported by a thin beam from the frame, fabricated in section 2-2, on the PDMS bracket.

Comment 5

Line 119: “In this process, as shown in Figure 6 (a)- 1, it starts from the situation where Face-up” should be “This process, shown in Figure 6 (a)- 1, starts from the situation where Face-up”

>> Thank you for the suggestion. We revised as mentioned.

Comment 6

Line 128: “Then, the ink is sintered to complete” to complete what? There is sthg missing in the sentence.

>> Thank you for the suggestion. We revised the manuscript as follows.

(Line 146)

Then, the ink is baked to complete sintering (Fig. 6 (a)-6).

Comment 7

Figure 8: I think it would be nice to say a few words on how these can be interpreted. What do you look at? The distance between peaks? The signal height? And what does it mean?

>>Thank you for the comment. We added the description as follows.

(Line 172)

In particular, two peaks (First peak and second peak) were observed on the wrist, and three peaks (first peak, second peak, and diastolic peak) were clearly measured at neck and ankle.